# Emotional Regulation and Adolescent Concussion: Overview and Role of Neuroimaging

**DOI:** 10.3390/ijerph20136274

**Published:** 2023-07-01

**Authors:** João Paulo Lima Santos, Meilin Jia-Richards, Anthony P. Kontos, Michael W. Collins, Amelia Versace

**Affiliations:** 1Department of Psychiatry, University of Pittsburgh School of Medicine, Pittsburgh, PA 15213, USA; jiarichardsm@upmc.edu (M.J.-R.); versacea@upmc.edu (A.V.); 2Department of Orthopaedic Surgery, UPMC Sports Concussion Program, University of Pittsburgh, Pittsburgh, PA 15213, USA; akontos@pitt.edu (A.P.K.); collinsmw@upmc.edu (M.W.C.)

**Keywords:** concussion, adolescence, post-concussion symptoms, emotion dysregulation, functional MRI, resting state MRI, diffusion MRI

## Abstract

Emotional dysregulation symptoms following a concussion are associated with an increased risk for emotional dysregulation disorders (e.g., depression and anxiety), especially in adolescents. However, predicting the emergence or worsening of emotional dysregulation symptoms after concussion and the extent to which this predates the onset of subsequent psychiatric morbidity after injury remains challenging. Although advanced neuroimaging techniques, such as functional magnetic resonance imaging and diffusion magnetic resonance imaging, have been used to detect and monitor concussion-related brain abnormalities in research settings, their clinical utility remains limited. In this narrative review, we have performed a comprehensive search of the available literature regarding emotional regulation, adolescent concussion, and advanced neuroimaging techniques in electronic databases (PubMed, Scopus, and Google Scholar). We highlight clinical evidence showing the heightened susceptibility of adolescents to experiencing emotional dysregulation symptoms following a concussion. Furthermore, we describe and provide empirical support for widely used magnetic resonance imaging modalities (i.e., functional and diffusion imaging), which are utilized to detect abnormalities in circuits responsible for emotional regulation. Additionally, we assess how these abnormalities relate to the emotional dysregulation symptoms often reported by adolescents post-injury. Yet, it remains to be determined if a progression of concussion-related abnormalities exists, especially in brain regions that undergo significant developmental changes during adolescence. We conclude that neuroimaging techniques hold potential as clinically useful tools for predicting and, ultimately, monitoring the treatment response to emotional dysregulation in adolescents following a concussion.

## 1. Introduction

Concussion, also known as mild traumatic brain injury (mTBI), is defined as a transitory disturbance in brain functioning due to complex pathophysiological processes induced by a traumatic event [1,2]. Sport- and recreation-related concussions represent the majority of these injuries in children and adolescents, affecting 1.9 million youth in the U.S. each year [3]. Diagnosing concussions and determining clinical recovery depend on an individual-centered approach. In this approach, identifying an ‘injury state’ depends on the presence of concussion symptoms that began immediately following the traumatic event. Common concussion symptoms include headache, fatigue, dizziness, nausea, memory difficulties, sleep disturbances, and changes in mood and personality [4]. Thus, the self-reporting (or caregiver-reporting) of these symptoms is a fundamental part of this diagnosis. This is complicated by the subjective nature of the symptom reporting (and, thus, by the possibility of patients over- or under-reporting their symptoms) and by the absence of objective diagnostic tests and/or biological markers of concussion [5]. As a result, a high percentage of concussions may go undiagnosed. In contrast, an equally high portion of those with new or preexisting psychiatric symptoms may be erroneously diagnosed with a concussion [6]. This becomes even more complicated in adolescents, where an undiagnosed or unresolved concussion might increase the risk of downstream consequences (e.g., prolonged symptoms, academic difficulties, and, less commonly, second impact syndrome or negative outcomes) [4,7] in developing brains.

On average, concussion symptoms in youth resolve within the first two weeks of injury [3,8,9]. However, in some instances, symptoms persist for longer, leading to Persistent Post-Concussion Symptoms (PPCS). Adolescents typically take longer (i.e., around four weeks) to recover from a concussion than children and adults [10,11,12], with adolescent girls taking the longest [13]. The longer recovery time can increase the impact of concussion on this age group. This may lead to persistent symptomatology, with approximately 21–35% of concussed adolescents developing PPCS [14,15,16]. PPCS in adolescents can adversely affect social functioning and academic performance [4] and are often associated with emotional dysregulation [17,18].

It is important to note that due to the rapid development of limbic regions prior to adolescence and the more gradual development of the prefrontal cortex throughout adolescence, this age period is considered a window of heightened vulnerability for psychiatric morbidity. The imbalance in maturity between limbic regions and the prefrontal cortex is thought to contribute to the typical difficulties in regulating emotions and behaviors that occur during adolescence (see Figure 1) [19]. Therefore, sustaining a concussion during adolescence may increase susceptibility to emotional dysregulation in this age group. Despite substantial progress in identifying the clinical outcomes of concussion [20,21,22], the complex pathophysiologic mechanisms underlying emotional dysregulation symptoms are poorly understood in both pediatric and adult samples.

Emotional regulation refers to the ability to monitor, evaluate, and adapt behaviors in response to emotional stimuli [23]. Emotions are patterns of perceptions, experiences, and reactions to challenging situations, and the dynamic cycle of perception, evaluation, and response changes continuously [24]. This adaptive function may help or harm someone, depending on how well emotions are regulated. Controversial among researchers is whether concussed patients develop emotional dysregulation symptoms as (1) a direct result of injury to brain regions involved in emotional regulation circuits, (2) a secondary effect of frustrations and/or uncertainty associated with their injury, or withdrawal from sports participation, and/or (3) because of a preexisting vulnerability.

Neuroimaging techniques, including computed tomography and magnetic resonance imaging, have been widely used to objectively characterize neurodevelopmental changes as they occur throughout the lifespan [25,26,27,28]. Neuroimaging has also been extensively used to detect brain abnormalities in concussed samples [29,30]. An emerging line of research further indicates that detecting abnormalities in emotional regulation regions may help explain anxiety symptoms in adolescents following concussion [29]. However, using these techniques is currently clinically indicated as neither a diagnostic nor a prognostic tool for concussion [30,31]. There are practical challenges to including advanced neuroimaging techniques in clinical practice [31], including (1) high costs for implementation and maintenance, (2) prolonged time to collect neuroimaging data that does not match current workflows, and (3) specialized training needed to properly interpret findings. Their use is also often inaccessible in clinical practice, contributing to their limited use in clinical settings. Nonetheless, neuroimaging techniques offer an objective way to identify neural correlates of concussion and/or predict outcomes in adolescents [30] and hold promise in further helping to disentangle the contribution of concussion-related brain abnormalities from changes associated with normative adolescent development, typical of this period. Demonstrating the utility of neuroimaging to identify abnormalities associated with emotional dysregulation in adolescent concussions may promote the use of neuroimaging in clinical practice.

The goal of this narrative review is to highlight current evidence on the utility of advanced neuroimaging techniques in investigating emotional dysregulation in adolescents following concussions. To this end, we provide a comprehensive narrative review of the available literature in electronic databases (PubMed, Scopus, and Google Scholar) using keywords such as concussion, post-concussion symptoms, emotional dysregulation, psychological symptoms, psychiatric symptoms, mood symptoms, depression, anxiety, neuroimaging, functional magnetic resonance imaging, and diffusion magnetic resonance imaging. An additional search was conducted using keywords such as children, adolescent, adolescence, youth, and adult to identify studies reporting findings in different age ranges between 2010 and 2023. To provide the reader with a conceptual framework, we will (1) review emotional dysregulation following concussion from a neurodevelopmental perspective; (2) provide an overview of how the employment of different neuroimaging modalities could enhance our understanding of the neuropathophysiological mechanisms driving emotional dysregulation in adolescent concussion; and (3) highlight neuroimaging findings linking concussion and emotional dysregulation from the existing literature. With this work, we seek to convey the breadth and depth of available research on the utility of advanced neuroimaging techniques in investigating emotional dysregulation in adolescent concussions. Additionally, we aim to identify research gaps and provide recommendations for future research in this area.

## 2. Emotional Dysregulation following Concussion

In the early—acute (<72 h) and sub-acute (72 h to 1 week)—stages of concussion, symptoms tend to be global in nature and reflect overall injury severity [32]. After the first week, symptoms typically cluster into more distinct clinical profiles—i.e., cognitive/fatigue, vestibular, ocular-motor, headache/migraine, cervical, and anxiety/mood [33]—with emotional dysregulation symptoms among the most challenging for clinicians and practitioners to recognize [34,35] and address [36]. With estimates ranging from 7% to 36%, emotional dysregulation symptoms in concussion are highly non-specific. These symptoms include anxiety, depression, affect lability, irritability, hostility, impulsivity, and personality changes. Prominent in mood and anxiety disorders, these symptoms are typically transient in concussion, resolving within one to two weeks of injury [32]. However, in some adolescents, these symptoms persist for longer (more than four weeks), with a significant minority experiencing emotional dysregulation symptoms up to one year post-injury [17,18,37,38]. For example, in a large (*N* = 14,765) sample of high school students in the United States, 36% of students with a concussion history reported persistent depressive symptoms for up to 12 months [35].

Research over the past decade has helped inform practitioners that depression [12,39,40], anxiety [40,41], posttraumatic stress disorder (PTSD) [42], substance use [43], and suicidal ideation or attempts [44,45,46] are prevalent in adolescents with a history of concussion. Nonetheless, estimates of psychiatric morbidity following concussion continue to vary (for recent reviews, see [47,48]), partly reflecting the challenges of identifying psychiatric symptoms in individuals without a preexisting psychiatric condition [49]. The heterogeneity of screening tools is another likely contributor to this variability, as these tools (e.g., Patient Health Questionnaire-9 and General Anxiety Disorder-7) [50,51] tend to be used in place of more rigorous structural clinical interviews or psychiatric rating scales in routine assessments of concussion [49]. Thus, while there has been substantial progress in understanding the clinical outcomes of concussion, it remains unclear whether we can identify patients at high risk for psychiatric morbidity following a concussion, especially adolescents.

## 3. Neural Correlates of Emotional Regulation

In the brain, ventro-limbic and dorso-limbic systems are involved in processing and regulating emotions (see Figure 2) [52]. The ventral system is primarily involved in coordinating information from the cortex, evaluating the salience of a stimulus, and eliciting an emotional state in response to the stimulus [53,54]. The amygdala plays a central role in the ventral system by evaluating emotional valence stimuli based on previously learned experiences [55,56,57]. The dorsal system is primarily involved in regulating emotional states and their associated behaviors [58,59]. The dorsal anterior cingulate gyrus is a hub within the executive functioning network [60], while the ventral striatum plays a key role in reward and reinforcement learning [61,62].

Within and between these systems, the subregions of the prefrontal cortex are directly and indirectly connected to other cortical and subcortical regions by major white matter tracts [59]. The inferior longitudinal fasciculus connects the occipital lobe with the medial temporal regions to attribute emotional salience to visual stimuli [63]. The uncinate fasciculus connects subcortical limbic regions with different subregions of the prefrontal cortex to regulate emotions and plan complex behaviors [64]. The cingulum bundle connects subcortical limbic regions with the cingulate cortex and is associated with emotional processing, emotional regulation, and executive functioning [64,65,66,67,68,69,70,71].

During adolescence, prefrontal cortical regions undergo significant structural and functional changes [72]. As adolescents transition from childhood to adulthood, the combination of pubertal and environmental factors (e.g., the rise of sex hormones, exposure to new experiences, social interactions, and acquisition of new skills) promote the strengthening or weakening of the connections within and between these systems [73]. The ability of the brain to reorganize its structure and function in response to environmental demand is referred to as brain plasticity [73]. The plastic changes that, during adolescence, occur in the prefrontal cortex support higher-order cognitive functions such as emotion regulation, decision-making, and social behavior [74]. The learning of new skills and the acquisition of new competence during this developmental period are key to adaptation [75]. While plasticity is beneficial for positive adaptation, it also increases the vulnerability of these systems to neurological injuries, such as concussions [76].

## 4. Role of Neuroimaging in Concussion

Concussions can damage the brain by either direct collision or the acceleration/deceleration forces associated with impact [77]. Direct collisions can cause diffuse damage due to the propagation of waves through the brain [78]. Biochemically, the effect of these forces on the brain is thought to activate a ‘neurometabolic cascade’ of events that, following the sheering/disruption of neuronal membranes, involves non-specific depolarization, the release of neurotransmitters (particularly excitatory amino acids), and the influx/efflux of ions such as calcium, magnesium, and potassium across the neuronal cytoplasmic membrane [78,79,80]. Concomitantly, there is a physiological decrease in cerebral blood flow [81]. The rapid and dynamic forces associated with the impact can cause profound geometric distortions and microscopic damage to the axon cytoskeleton of white matter fibers (diffuse axonal injury). This type of injury is usually associated with axonal dysfunction [82]. Frontal, temporal, and occipital midline brain regions are particularly vulnerable to acceleration and deceleration forces [83].

Neuroimaging procedures offer a way to examine the pathophysiologic processes of concussion in vivo. Quick and capable of detecting a broad spectrum of pathologies, computed tomography is the current standard neuroimaging procedure for brain injuries on the day of injury. However, patients with concussion typically undergo neither computed tomography nor ‘conventional’ magnetic resonance imaging as the diagnostic capacity of these procedures for mild forms of brain injury is limited [84]. In fact, computed tomography is not suitable for the detection of developmental changes occurring in the adolescent brain or pathophysiological processes characteristic of acute, subacute, or chronic stages of concussion [31,85]. These limitations significantly restrict the clinical utility of this technique in concussion. Current guidelines for adults and pediatrics (e.g., the American Medical Society of Sports Medicine) recommend clinical neuroimaging only when there is concern about intracranial hemorrhage [31]. Magnetic resonance imaging is also widely used in clinical settings as an essential tool for diagnosing and monitoring a wide range of medical conditions. Technological advances in magnetic resonance imaging have led to improvements in image quality through the use of higher magnetic fields (3 Tesla and above) and a decrease in scanning time. Techniques such as GeneRalized Autocalibrating Partially Parallel Acquisitions—GRAPPA, SENsitivity Encoding—SENSE, and/or Simultaneous Multi-Slice Imaging—SMS enable the acquisition of high-quality images in a relatively short period of time. Of the magnetic resonance imaging modalities, functional and diffusion magnetic resonance imaging have been widely used to characterize different properties of the brain functioning and structure, ranging from normative brain development to medical conditions [86,87,88,89].

Over the past fifteen years, the employment of advanced magnetic resonance imaging methods has promoted a better understanding of the pathophysiologic mechanisms of concussion. Their use in longitudinal designs is likely to shed light on important mechanisms underpinning the acute, sub-acute, and chronic stages of concussion. For example, in adolescence, these approaches are well-suited to determine if a progression of concussion-related abnormalities exists in chronic stages of concussion after disentangling the contribution of the neurodevelopmental changes typical of these ages.

### 4.1. Functional Magnetic Resonance Imaging

Broadly, functional magnetic resonance imaging measures the hemodynamic response, the Blood-oxygen-level-dependent (BOLD) signal, as a proxy of neural activity. This activity corresponds to the oxygen consumption of firing neurons, and it can be assessed either during rest (resting state magnetic resonance imaging) or during task performance (task magnetic resonance imaging) [90,91]. In addition, these methods allow for identifying the temporal correlation between brain regions and quantifying their functional integration or connectivity [90,91].

When analyzing resting-state magnetic resonance imaging, researchers generally focus on functional segregation or functional integration across networks [92,93,94,95,96]. Different statistical approaches have been proposed to investigate these aspects of brain activity. Functional segregation examines regional characteristics of the brain and divides these regions based on their functionality [92,93,94,95,96]. Proposed approaches involve identifying (1) the Amplitude of Low Frequency Fluctuations (ALFF) for individual regions or (2) the Regional Homogeneity (ReHo) or synchronization between regions and their nearest neighbors [92,93,94,95,96]. Functional integration measures the connectivity and synchrony between different regions of the brain [92,93,94,95,96]. This connectivity can be direct (a structural pathway connecting two regions) or indirect (a connection between two regions is mediated by another region) [92,93,94,95,96]. Common approaches to assessing functional integration are Independent Component Analysis (ICA) and Graph theory analysis [92,93,94,95,96]. ICA involves separating the functional signal into independent functional networks based on synchronized BOLD activity and measuring the connectivity within such networks [92,93,94,95,96]. Graph theory analysis models the brain as a network of nodes and edges [92,93,94,95,96]. Nodes indicate the brain regions and the edges indicate the connection between these brain regions [92,93,94,95,96]. Within these models, several features are measured, including path length (number of edges), clustering (local neighborhood connectivity), and small-world (connectivity between nodes when most of them are not directly connected but can be reached through a few connections) [92,93,94,95,96]. In concussion, these methods are suitable for identifying region-specific or network-based changes following injury, how brain networks adapt to the functional changes associated with concussion, and whether changes in connectivity patterns are implicated in the presence or persistency of symptoms, including emotional dysregulation symptoms. Resting state studies in adolescents who sustained a concussion have shown hyperconnectivity and hypoconnectivity compared to controls, depending on the regions studied [97,98,99,100]. Hyperconnectivity has been found within posterior regions (e.g., precuneus and cerebellum) [99] and between the salience network and cerebellum [98]. Hypoconnectivity has been found within anterior regions (inferior and middle frontal gyrus) [99], between the dorsal attention network and inferior frontal gyrus [100], within the default mode network (involved with episodic memory and emotional processing) [97], and between the salience network and the thalamus [98]. However, the correlation between clinical measures and resting state abnormalities has yet to be extensively studied.

Task magnetic resonance imaging maps the response of the brain to tasks administered during imaging collection [101]. Different tasks have been used in humans to engage functionally distinct brain regions while performing a given task. These tasks include the stop signal, which assesses response inhibition by measuring the reaction time to stop and go trials [102]. The monetary incentive evaluates the anticipation and feedback processing of rewards that depend on task performance [103]. Additionally, the emotional face N-back task examines attentional control during a memory task while presenting emotional distractors such as happy, fearful, and neutral faces [104]. In essence, the increased bold signal change elicited by a task (hyperactivation) is thought to reflect neural activity associated with the recruitment of greater brain resources or a compensatory response required to maintain task performance. On the other hand, decreased bold signal change elicited by the same task (hypoactivation) is thought to reflect difficulties in allocating resources to perform this task [101,105]. In concussion, this approach allows us to understand how the brain functionally responds to the direct or indirect effects (e.g., neurometabolic changes) of injury and which patterns of brain activity may help explain the cognitive and emotional challenges adolescents face following injury. Task magnetic resonance imaging findings suggest that those with concussion need to recruit more brain resources to sustain working memory [106,107,108,109,110,111,112,113,114] and attention [115,116] when compared to controls. In adolescents with concussion, poorer working memory is associated with reduced cortical activation, especially in the dorsolateral prefrontal cortex [117,118,119].

### 4.2. Diffusion Magnetic Resonance Imaging

On the other hand, diffusion magnetic resonance imaging provides indices of tissue microarchitecture by measuring the diffusion imaging properties of the water molecules in the brain tissue [120,121,122]. This technique enables the study of structural connectivity between brain regions. Fractional anisotropy is the most common metric used in diffusion magnetic resonance imaging, indexing the structural collinearity of the fibers and/or their integrity—e.g., damage to axon membranes or myelin sheaths [121,122]. It reflects the preponderance of the water motility along the principal diffusion direction (axial diffusivity) over secondary diffusion directions that are transverse to the axial diffusivity (i.e., radial diffusivity) within a voxel.

Advances in diffusion magnetic resonance imaging have increased our ability to characterize micro- and macro-structural properties of different brain tissues and detect abnormalities across different medical conditions (e.g., diffuse axonal injury, transitory ischemic attack, stroke) [123,124]. State-of-the-art diffusion imaging protocols collect a high number of diffusion gradient directions (i.e., High Angular Resolution Diffusion Imaging; HARDI) that allow resolving the crossing fiber issue, for which the tensor model remains agnostic. Recent recommendations suggest including multiple b-values (i.e., multishell diffusion imaging [86,125] and Diffusional Kurtosis Imaging; DKI) that, by using multiple concentric shells, can either inform on the fine-grained micro-structural properties of the brain tissues or quantify the deviation of diffusion imaging properties from Gaussian behavior. Specifically, by using different b values, it is possible to better account for the effect of noise in the diffusion imaging data [87,126,127]. These advanced protocols allow for the employment of multi-tensor models and cutting-edge mathematical algorithms, such as Multishell Multi-Tissue Constrained Spherical Deconvolution (MSMT-CSD) and Neurite Density Dispersion Imaging (NODDI). MSMT-CSD is a diffusion imaging model that maximizes the precision of the fiber orientation distribution function in white matter voxels and minimizes partial volume bias in voxels containing gray matter and/or corticospinal fluid [128,129,130]. NODDI allows for the compartmentalization of the diffusion imaging signal in intra- (i.e., axons and dendrites) and extra- (i.e., glia, cerebrospinal fluid) neuritic spaces, leading to estimates of the microstructural complexity of different tissues in the brain [131]. Notably, NODDI measures have been shown to be more sensitive than standard diffusion tensor imaging measures (e.g., fractional anisotropy) to structural-developmental changes in normative [25,132,133] and clinical [87,88,89,134,135,136,137] samples. Specific NODDI metrics include the free-water isotropic volume fraction (FISO), reflecting the free water volume fraction; the Neurite Density Index (NDI), reflecting the intra-neurite density; and the Orientation Dispersion Index (ODI), reflecting the angular dispersion of neurites.

Studies using diffusion magnetic resonance imaging have found concussion-related microstructural abnormalities in several brain regions [138,139,140], including the corpus callosum [141,142,143,144,145,146,147,148,149,150,151,152,153,154,155,156,157,158,159,160,161,162,163], internal capsule/anterior thalamic radiation [147,149,153,156,158,159,162,163,164,165,166], and centrum semiovalis/superior longitudinal fasciculus [138,147,158,160,162]. Lower fractional anisotropy in the inferior longitudinal fasciculus and the inferior fronto-occipital fasciculus has been associated with longer recovery time in adolescent concussion [167]. This is likely due to a lack of long associative tracts and their integrative functions. In adolescents, diffusion magnetic resonance imaging abnormalities have been detected up to a month post-injury despite improvements in cognitive ability, suggesting that these abnormalities may persist even after full clinical recovery [168]. Yet, the long-term clinical implications of these findings remain unclear. Few studies have applied NODDI to study concussion in adolescents. One study has shown an association between lower NDI in the corpus callosum and increased brain-injury blood markers (e.g., tau) in adolescent concussions [169]. Another NODDI study has shown that concussion, when followed by poor quality sleep, is associated with decreased NDI in several white matter tracts such as the cingulum bundle, optic radiation, striato-fronto-orbital tract, and superior longitudinal fasciculus I. Furthermore, the study found that decreased NDI in these tracts is associated with increased postconcussive symptom severity in adolescents [170].

### 4.3. Other Neuroimaging Modalities

While functional and diffusion imaging have been mostly used to document brain abnormalities in concussion, other modalities (e.g., susceptibility-weighted imaging, magnetic resonance spectroscopy, and dynamic susceptibility contrast) have also been used.

Susceptibility-weighted imaging is susceptible to the parametric properties of different blood products of micro- and macro-hemorrhage (e.g., deoxyhemoglobin and hemosiderin). In concussion, employing this technique allows the detection of micro-hemorrhagic lesions that would otherwise go unnoticed (so-called ‘non-hemorrhagic shearing injury’) [171]. Some studies have shown a relationship between the number and volume of acute hemorrhages and clinical outcomes in moderate/severe forms of traumatic brain injury. However, susceptibility-weighted imaging results are inconclusive for mild traumatic brain injury/concussion (for recent reviews, see [30,171,172,173,174]).

Magnetic resonance spectroscopy is a metabolic imaging method that detects and quantifies metabolites (e.g., N-Acetyl Aspartate, Choline, and Creatine) in the brain tissue [175]. In concussion, magnetic resonance spectroscopy allows for studying neurometabolic changes associated with concussion-related lesions [172,176]. Low levels of n-acetyl aspartate and high levels of choline have been interpreted as neuronal damage/stress and glia abnormalities, respectively, in concussion. The magnitude of the changes has also been associated with the severity of symptoms and history of previous concussions. In contrast, the persistency of abnormalities after acute/subacute periods has been associated with persistent post-concussion symptoms [172].

## 5. Magnetic Resonance Imaging Findings Linking Concussion and Emotional Regulation from the Existing Literature

There has been a paucity of studies examining the structural and functional neural changes accompanying emotional dysregulation in adolescent concussion, especially as to whether detecting concussion-related abnormalities in the brain relates to psychiatric vulnerability following concussion (Table 1). Important evidence of the relationship between brain structure and emotional dysregulation in concussion comes from animal studies [177,178].

Animal studies have shown increased anxiety-related behaviors after a concussion, as measured by the locomotor activity of concussed rats in a maze. These behaviors were associated with increased neuronal excitability of the amygdala, as demonstrated by decreased inhibitory synaptic transmission [177]. Another study focusing on the amygdala examined the freezing behavior of mice under a neutral condition in a fear conditioning paradigm. These mice showed decreased fear response after concussion and decreased amygdala activity in voltage-sensitive dye imaging [178]. Overall, these findings suggest that amygdala dysfunction after injury may explain part of the pathophysiological processes of concussion. Furthermore, it possibly represents the neural underpinnings of anxiety-related behaviors observed in animal models of concussion.

In the human literature, children and adolescents appear to have abnormal neural responses to emotional stimuli after injury [183,184]. Specifically, using functional magnetic resonance imaging, studies have found blunted amygdala responding to emotional stimuli in 14–18-year-old adolescents with a concussion, with a stronger effect in those with persistent symptomatology [183]. In addition to amygdala dysfunction, concussed adolescents—compared to non-concussed adolescents— have also shown a wider spread of frontal lobe activation when requested to process emotional stimuli during an inhibitory control task. This suggests needing to compensate more when emotional information interferes with an inhibitory control task [184]. Amygdala dysfunction and abnormal communication between the amygdala and frontal lobes have been consistently implicated in the pathophysiology of both depression and anxiety disorders [111,188,189,190,191,192,193]. Thus, concussion-related lesions affecting these regions may contribute to the emotional dysregulation often reported after a concussion.

Evidence from adults suggests that white matter abnormalities in those who developed depressive symptoms following concussion do not differ from those found in adults with depression alone [186]. Specifically, when comparing adults with depression to adults with concussion and depression, Maller et al. found that both groups had reduced collinearity in white matter tracts that are heavily involved in regulating emotions. These tracts include the forceps minor and superior longitudinal fasciculus [186]. Diffusion imaging abnormalities in emotional regulation circuits have been associated with new-onset depression following concussion [194], similar to those reported in depression [195,196].

To our knowledge, few studies have examined the relationship between the concussed brain and the emergence of depression or anxiety in children and adolescents. Max, Keatley [181] found evidence for a relationship between concussion-related abnormalities in brain regions involved in emotional regulation and the onset of subsequent psychiatric morbidity. Specifically, they found fronto-temporal lesions in gray (inferior frontal gyrus) and white (frontal and temporal) matter regions in a sample of children and adolescents (*n* = 15) who developed full or subclinical forms of depression or anxiety six months post-injury. Most of them (60%) had no prior psychiatric history [181]. Our recent work with concussed adolescents suggests that sex and pubertal status may also play a role [29]. Compared to non-concussed adolescents, concussed adolescents with more advanced pubertal maturation had lower neurite density index (a NODDI metric that represents water diffusivity in the intraneuritic space) [131] in the cingulum bundle and forceps minor [29]. These white matter tracts are involved in emotional regulation [64]. A lower neurite density index in these tracts was also associated with higher levels of anxiety shortly after the concussion, particularly in girls [29]. These findings suggest that girls at a more advanced stage of puberty might be at greater risk for developing psychological problems from concussion compared to boys at a similar pubertal stage or girls at a less mature stage of puberty. What remains unknown is if these structural abnormalities are associated with transient symptomatology (e.g., acute symptoms of emotional dysregulation that resolve within a few weeks) or if they relate to an increased vulnerability for psychiatric morbidity later in life.

## 6. Factors Contributing to a Knowledge Gap in the Current Literature

As noted above, adolescence is a sensitive developmental window characterized by structural and functional changes reflecting brain plasticity, and injury may induce a risk for psychiatric morbidity. A key aspect that needs further investigation is whether a concussion in the adolescent brain can affect the establishment of connections that support emotion regulation. This may lead to heightened vulnerability to anxiety and depression later in life. As noted above, adolescence is a sensitive developmental window characterized by structural and functional changes reflecting brain plasticity, and injury may induce risk. However, the lack of longitudinal studies, the paucity of studies in youth, and heterogeneity due to a lack of standardized protocols across studies contribute to this knowledge gap. These factors limit the interpretability of the existing findings and their clinical and translational relevance [197].

To properly address this question, longitudinal neuroimaging studies are needed to disentangle the effects of concussion on the adolescent brain from those associated with typical development or those related to preexisting psychiatric vulnerability (e.g., history of trauma, familial history) and morbidity (e.g., neural correlates of a psychiatric illness and/or the effects of psychotropic medications). Altogether, the literature reviewed in this section provides support for the employment of advanced neuroimaging protocols as a tool to determine the effect of concussions on the developing brain. Moreover, it further determines if a progression of concussion-related lesions in brain regions involved in emotional regulation relates to a heightened vulnerability for psychiatric morbidity in adolescents who sustain a concussion. However, the lack of longitudinal studies, the paucity of studies in youth, and heterogeneity due to a lack of standardized protocols across studies limit the interpretability of these findings and their clinical translational relevance [197].

## 7. Clinical Relevance

Understanding the neural mechanisms of emotional dysregulation following concussion in adolescents may promote early identification of at-risk adolescents. Specifically, based on the networks structurally and/or functionally affected by concussion, advanced neuroimaging techniques may help (1) assess risk for persistent emotional dysregulation symptoms and psychiatric morbidity, (2) tailor individualized treatment approaches (e.g., cognitive/behavioral therapies), (3) track recovery progress, and (4) inform decisions regarding return to activities. The identification of specific regions implicated in emotional dysregulation following concussion in adolescents might also pave the way for the development and inclusion of novel personalized treatments in clinical practice. One such example is neurofeedback, directed at modulating the functioning of identified targeted regions that may be key for recovery in adolescent concussion. Neurofeedback is a form of biofeedback technique that aims to train individuals to consciously control their brain activity [198]. Currently, there is limited evidence of the utility of neurofeedback in treating concussions [199,200]. However, this approach has been successfully implemented to strengthen emotional regulation networks [201] and treat emotional dysregulation disorders such as depression [202] in adolescents.

## 8. Recommendations for Future Research

There is a paucity of studies directly examining the relationship between concussion-related brain abnormalities and heightened vulnerability to emotional dysregulation in adolescents. The identification of structural and/or functional abnormalities in emotional regulation circuits may serve as biomarkers of emotional dysregulation in adolescent concussion and promote the identification of those adolescents who are most at risk for psychiatric morbidity following concussion. To achieve this, future studies should employ advanced neuroimaging techniques in longitudinal designs that allow for the identification of functional and structural changes in regions and networks involved with emotional regulation and the effects of environmental factors (e.g., sleep and physical activities) on brain recovery post-concussion. However, there are important steps to consider before contemplating their implementation into clinical practice: (1) Standardize neuroimaging protocols and analytic methods to ensure consistency and facilitate comparison across different sites; (2) establish data sharing protocols to promote replication of findings incorporating different socio–demographic factors (e.g., race, ethnicity, sex, income, and parent education); (3) foster communication between researchers and clinical providers to facilitate the translational aspect of concussion neuroimaging research; and (4) perform validation studies that address the clinical relevance and cost-effectiveness of these techniques in clinical settings.

## 9. Conclusions

Although variability in study design, age ranges, sample sizes, and post-injury time points limits conclusions, the literature reviewed above supports the employment of advanced magnetic resonance imaging techniques in adolescent concussions. These techniques have the potential to achieve several key objectives: (1) promote a better understanding of the different pathophysiologic mechanisms of acute, subacute, and chronic stages of concussion; (2) discern between structural and functional abnormalities associated with a concussion from those associated with normative development or preexisting psychopathology, which may overlap (3) foster early identification of risk for psychiatric outcomes; and (4) serve as a prognosis tool for returning to activities. These can be instrumental in improving clinical decision-making and fostering the identification of early intervention strategies aimed at strengthening emotional and behavioral regulation strategies in at-risk groups.

## Figures and Tables

**Figure 1 ijerph-20-06274-f001:**
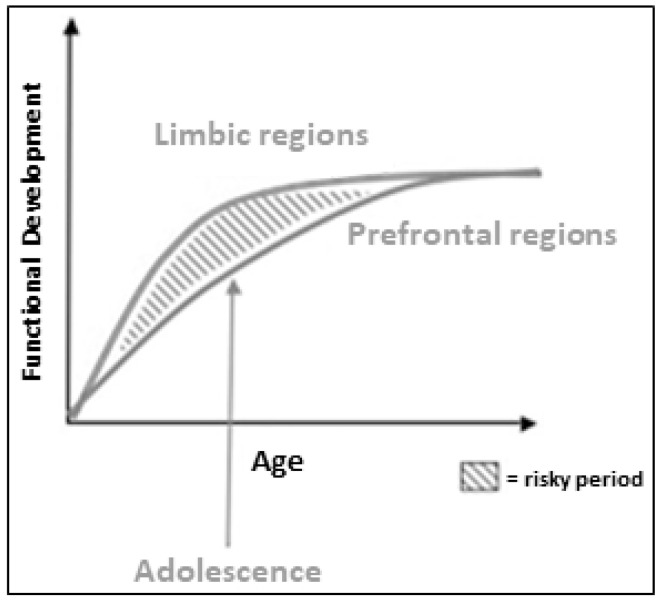
Developmental curves of neural circuitries. This figure displays the differences between developmental trajectories of limbic systems and prefrontal control regions over time (Figure reproduced with permission from Ref [19]. Copyright 2007 by Elsevier Inc.).

**Figure 2 ijerph-20-06274-f002:**
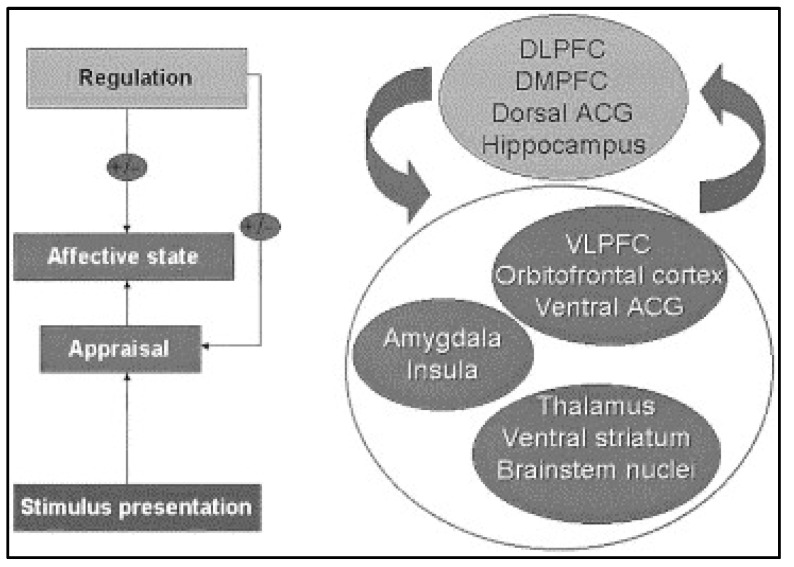
Schematic diagram of emotion perception. This figure displays the brain regions important for the processes underlying emotion perception. Abbreviations: DLPFC, dorsolateral prefrontal cortex; DMPFC, dorsomedial prefrontal cortex; ACG, anterior cingulate gyrus; VLPFC, ventrolateral prefrontal cortex (Figure reproduced with permission from Ref. [52]. Copyright 2003 by Society of Biological Psychiatry).

**Table 1 ijerph-20-06274-t001:** Summary of Neuroimaging Studies on Concussion and Emotional Regulation.

Author(s)	Year	Title	Sample	Modality	Findings
Dégeilh et al. [179]	2022	Social problems and brain structure trajectories following pediatric mild traumatic brain injury	Children with (*n* = 224) and without (*n* = 5736) a history of concussion; assessed at 9–10 and 11–12 years old	sMRI	Youths with a suspected history of concussions had higher levels of social problems than youth without concussions. No differences were found in cortical thickness between youth with and without concussion.
Lopez et al. [34]	2022	Association between mild traumatic brain injury, brain structure, and mental health outcomes in the Adolescent Brain Cognitive Development Study	Youth with concussion (*n* = 199) and possible concussion (*n* = 527); assessed at 9–10, 10–11, and 11–12 years old	sMRI	Youths with a suspected history of concussion had higher levels of emotional and behavioral problems than youths without. Measures of brain structure did not mediate the relationship between concussion and increased mental health problems.
Vasa et al. [180]	2015	Prevalence and predictors of affective lability after paediatric traumatic brain injury	Children and adolescents with TBI; *N* = 97; 4–19 years	sMRI	Orbitofrontal cortex injury predicted emotional and behavioral problems 12 months post-injury.
Max et al. [181]	2012	Depression in children and adolescents in the first 6 months after traumatic brain injury	Youth with a history of TBI (*N* = 177; 69% with mild, 18% with moderate, 54% with severe TBI); 5–14 years old	sMRI	11% of youth had new-onset depression following injury; left IFG and right frontal white matter lesions were associated with new-onset depression.
Wilde et al. [182]	2012	Longitudinal changes in cortical thickness in children after traumatic brain injury and their relation to behavioral regulation and emotional control	Children and adolescents; *n* = 20 with TBI, *n* = 21 with orthopedic injury; 7–18 years	sMRI	Youth with TBI showed cortical thinning over time; thinning in the right medial frontal and right anterior cingulate gyrus of youth with TBI associated with poorer emotional control over time; thinning in the medial left frontal lobe in youth with TBI associated with poorer behavioral control over time.
Stein et al. [118]	2021	Changes in working memory-related cortical responses following pediatric mild traumatic brain injury: A longitudinal fMRI study	Youth with concussion; *N* = 29; 8–18 years	fMRI	Greater activation of DLPFC and DMN was associated with improved working memory performance and fewer emotional and behavioral problems between 1-month and 2-month post-injury.
Bohorquez-Montoya et al. [183]	2020	Amygdala response to emotional faces in adolescents with persistent post-concussion symptoms	Adolescents with concussion with PPCS (*n* = 23) and without PPCS (*n* = 13), and healthy controls (*n* = 15); 14–18 years old	fMRI	Adolescents with PPCS had a blunted amygdala response to emotional stimuli compared to healthy controls and, to a lesser extent, adolescents without PPCS.
Ho et al. [184]	2018	An Emotional Go/No-Go fMRI study in adolescents with depressive symptoms following concussion	Youth with concussion; *N* = 30; 10–17 years old	fMRI	Youth with high levels of depression had reduced frontal cortex activity in response to negative emotional stimuli compared to those with low-to-moderate levels of depression.
Santos et al. [29]	2022	The role of puberty and sex on brain structure in adolescents with anxiety following concussion	Adolescents with recent concussion diagnosis (*n* = 55) and matched controls (*n* = 50); 12–17 years	dMRI	Compared to controls, adolescents with concussion with more advanced pubertal maturation had a lower neurite density index in cingulum bundle and forceps minor. Lower neurite density index was associated with higher levels of anxiety, especially in girls.
Alhilali et al. [185]	2015	Evaluation of white matter injury patterns underlying neuropsychiatric symptoms after mild traumatic brain injury	Adults, children, and adolescents with concussion; *n* = 38 with irritability; *n* = 32 with depression, *n* = 18 with anxiety, *n* = 29 controls; 10–47 years	dMRI	Compared to controls, those with depression had reduced FA in the right NAc, anterior limb of the internal capsule, and SLF; those with anxiety had reduced FA in the cerebellar vermis; no significant differences were found in those with irritability; in those with depression, NAc FA was negatively correlated with recovery time.
Maller et al. [186]	2014	The (Eigen) value of diffusion tensor imaging to investigate depression after traumatic brain injury	Adults, *n* = 26 with depression only, *n* = 12 with concussion only, *n* = 15 with new-onset depression and concussion; *n* = 25 healthy controls	dMRI	Reduced axial diffusivity in DLPFC, CC, and NAc white matter tracts in those with new-onset depression.
Rao et al. [187]	2012	Diffusion tensor imaging atlas-based analyses in major depression after mild traumatic brain injury	Adults with concussion, *n* = 21; ~36 years	dMRI	Reduced FA in fronto-temporal white matter was associated with new-onset depression 12 months post-concussion.

Note. sMRI = structural magnetic resonance imaging; fMRI = functional MRI; dMRI = diffusion MRI; TBI = traumatic brain injury; FA = fractional anisotropy; CC = corpus callosum; ILF = inferior longitudinal fasciculus; IFOF = inferior fronto-occipital fasciculus; SLF = superior longitudinal fasciculus; ACC = anterior cingulate cortex; dACC = dorsal ACC; PFC = prefrontal cortex; DLPFC = dorsolateral PFC; VLPFC = ventrolateral PFC; NAc = nucleus accumbens; DMN = default mode network.

## Data Availability

Not applicable.

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
