# Peer review of "Emotional Regulation and Adolescent Concussion: Overview and Role of Neuroimaging"

_ijerph, 2023, doi:10.3390/ijerph20136274_

Round 1
Reviewer 1 Report
An interesting paper on the potential utility of neuroimaging techniques in the the identification of emotional dysregulation in adolescent patients with mild traumatic brain injury.
1. A couple of minor typographical errors- use correct abbreviation in line 60 (PPCS not PCCS). Spell out the abbreviations in line 233.
2. Color images to illustrate the neuroimaging findings described for each neuroimaging technique would be helpful if available
3. The authors conclude that the data to support their recommendation for the use of neuroimaging to identify adolescents with TBI at risk for emotional dysregulation is limited. Unclear, therefore, what the real life beneficial practice implications of the information presented are.
4. The authors additionally do not discuss the practical challenges to their recommendations for example cost of the different neuroimaging modalities, limited availability of these imaging modalities, patient variability with regards to injuries, implications of underlying pre-injury neuropsychiatric/emotional disorders, etc.
5. The authors should give a reference for the brain stimulation they introduce in line 539 or remove it since this is not standard practice in adolescent TBI management.
A couple of minor typographical errors- use correct abbreviation in line 60 (PPCS not PCCS). Spell out the abbreviations in line 233.
Reviewer 2 Report
Response to authors
Emotional Regulation and Adolescent Concussion: Overview 2 and Role of Neuroimaging
I would like to express my gratitude for the opportunity to read this research. The recommendations in this document are intended to help you improve your work. Here are a few small points to consider:
Abstract
Please review abstract thoroughly and added the methods that used in this article.
Content
You have three purposes: 1.) to review emotional dysregulation following concussion from a neurodevelopmental perspective, 2.) to provide an overview of how the employment of different neuroimaging modalities could enhance our understanding of the neuropathophysiological mechanisms driving emotional dysregulation in adolescent concussion, and 3.) to highlight neuroimaging findings linking concussion and emotional dysregulation from the existing literature. You should explain the method to answer these purposes.
Line 119 point 2, Line 154 point 3, Line 200 point 4. Line 251 point 4.1 until point 5: I propose authors write the hypotheses of the study before discussing this point.
Result and Discussion for this article is not clear enough because the author did not explain about type of the study. Is it analysis review or qualitative research?
Conclusion
Line 515: conclusions are adjusted to the introduction and research objectives. Please also check that you have used the references according to these guidelines.
Reviewer 3 Report
Thank you for the opportunity to review this manuscript, it's an important topic. I have just a few comments:
Page 2, lines 49-52: can the authors provide a reference that second impact syndrome (a very serious but rare outcome of concussion) is more of a risk in adolescents? I agree adolescents are at increased risk for complications for several reasons but second impact syndrome, given its rarity, is not one that stands out – please add additional, more common, reasons for the increased risk in adolescents and include second impact syndrome if in fact that is one where adolescents are at higher risk.
Page 2, lines 53-54: add “on average” to two weeks, clarify if that statement is for adults only, or rephrase that full recovery can take a few weeks.
Page 2, line 85: add “and/or” – these possibilities are not mutually exclusive
Page 4, lines 132-137: Are these estimates of mood symptoms independent of other symptoms or comorbid with them? For example, what proportion of the 36% reporting depressive symptoms had other possibly contributory symptoms, like pain, etc? did these estimates account for premorbid mood problems?
Table 1: How were these papers identified? Which got excluded from this list?
Conclusions: The paper raises several important questions and concerns, and highlights many unknowns about the brain-based (function and structure) contributions about emotion regulation after concussion in adolescents. As a reader, I want a little more about what the authors have identified regarding future directions, specifically what methodological recommendations can they make based on the literature they've reviewed? what are some research priorities? how might identifying brain-based biomarkers improve intervention beyond standard of care interventions for moos post concussion? is it possible to isolate emotional regulation from other nervous system regulatory processes? how so and what does this mean in terms of intervention?
Round 2
Reviewer 2 Report
Dear Author,
Thank you for the explanation, and your manuscript deserves to be published,
Best Regard,